# Surface Modification of 3D Printed PLA/Halloysite Composite Scaffolds with Antibacterial and Osteogenic Capabilities

**Yangyang Luo** [1], **Ahmed Humayun** [1] **and David K. Mills** [1,2,*]

[1] Center for Biomedical Engineering and Rehabilitation Science, Louisiana Tech University, Ruston, LA 71272, USA; yangyang317luo@gmail.com (Y.L.); ah.humayun@gmail.com (A.H.)

[2] School of Biological Sciences, Louisiana Tech University, Ruston, LA 71272, USA

\* Correspondence: dkmills@latech.edu; Tel.: +1-318-257-2640

**Abstract:** Three-dimensional (3D) printing techniques have received considerable focus in the area of bone engineering due to its precise control in the fabrication of complex structures with customizable shapes, internal and external architectures, mechanical strength, and bioactivity. In this study, we design a new composition biomaterial consisting of polylactic acid (PLA), and halloysite nanotubes (HNTs) loaded with zinc nanoparticles (PLA+H+Zn). The hydrophobic surface of the 3D printed scaffold was coated with two layers of fetal bovine serum (FBS) on the sides and one layer of NaOH in the middle. Additionally, a layer of gentamicin was coated on the outermost layer against bacterial infection. Scaffolds were cultured in standard cell culture medium without the addition of osteogenic medium. This surface modification strategy improved material hydrophilicity and enhanced cell adhesion. Pre-osteoblasts cultured on these scaffolds differentiated into osteoblasts and proceeded to produce a type I collagen matrix and subsequent calcium deposition. The 3D printed scaffolds formed from this composition possessed high mechanical strength and showed an osteoinductive potential. Furthermore, the external coating of antibiotics not only preserved the previous osteogenic properties of the 3D scaffold but also significantly reduced bacterial growth. Our surface modification model enabled the fabrication of a material surface that was hydrophilic and antibacterial, simultaneously, with an osteogenic property. The designed PLA+H+Zn may be a viable candidate for the fabrication of customized bone implants.

**Keywords:** 3D printing; antibiotics; bone regeneration; composites; halloysite; zinc

## 1. Introduction

According to the reports of the National Ambulatory Medical Care Survey and American Academy of Orthopedic Surgeons, about 6.8 million patients ask for medical therapy due do orthopedic problems every year, and more than two million bone grafting procedures are performed annually [1]. Autografts are considered the gold standard for bone repair because of their excellent properties in osteoconduction, osteoinduction, and osteogenesis [2]; however, autografts have many limitations. These include their limited availability, the requirement for a surgical incision to obtain the graft that carries the extra risks of hematoma, infection, and additional pain [3]. Allografts are another source for orthopedic implants, and nearly one-third of all bone grafts used in North America are allografts [4]. However, allografts are osteoconductive but with reduced osteoinductivity, which increases the risk of nonunion in fracture repair, and there is a risk of infection [5,6]. In addition, the supply of allografts is limited by the long pretreatment process. For the reasons outlined above, a new method for fabricating a bioengineered bone graft with the proper mechanical properties, osteoconductivity, and osteogenic abilities is highly desired.

Bone implants can be produced through a variety of methods including salt leaching [7], chemical/gas foaming [8], freeze-drying [9], and sintering [10]. However, pore size, pore distribution, porosity, and pore interconnectivity cannot be precisely controlled with these approaches [11]. Bone is a porous tissue with numerous interconnected pores that permit cell migration and proliferation, as well as vascularization [12,13]. Therefore, an osteogenic scaffold should mimic bone morphology, structure, and function in order to ensure its integration with the native tissue. Three-dimensional (3D) printing technology has received considerable attention for tissue regeneration due to its superiority in the fabrication of complicated structures with tailored shapes, internal and external architecture, pre-designed microstructure, mechanical strength, and bioactivity, which can effectively mimic native tissues. With the use of osteogenic biomaterials and computer-aided design, 3D printing technology can generate a customized structure with desired features that can improve bone integration and the restoration of tissue function [14].

Hybrid materials with tunable properties have been explored in 3D printing [14–16]. Polylactic acid (PLA) is a popular material used for 3D printing medical devices. It is a thermoplastic polymer that is derived from fermented corn starch, cassava starch, or sugarcane [17]. It is an eco-friendly bioplastic as it is entirely biodegradable and consists of renewable raw materials. This material exhibits high tensile strength, low elongation, and high modulus, which enables it to be a suitable candidate for load-bearing applications, such as orthopedic fixation and sutures [17].

In this study, PLA was used to fabricate porous scaffolds through 3D printing. According to previous reports, large pore size and high porosity are key factors in producing an osteogenic response [12,13,18]. In addition, a recent study reported that titanium implants with an average pore size of 600 μm exhibited an earlier and high fixation ability and rapid bone ingrowth comparing with implants with average pore size of 300 and 900 μm [19]. Therefore, we designed the scaffold with an average pore size of 600 μm and 60% porosity. PLA is a versatile, biodegradable, and Food and Drug Administration (FDA) proved biomaterial [17], but its surface is hydrophobic and, therefore significantly reduces cell adhesion. Accordingly, surface modification to enhance its cell supportive material properties can be achieved through the addition of micro- and nanoparticles [14], fibers [16], or fabrication of nanocomposites [15]. Halloysite is an aluminosilicate clay $Al_2O_3 \cdot 2SiO_2 \cdot nH_2O$, with a naturally aluminosilicate rolling numerous nanotubes formed. Those nanotubes have an average length of 0.5–2 μm and diameter of 50–80 nm, in addition these nanotubes have hollow lumen with an average diameter of 10–15 nm [20]. Halloysite nanotubes (HNTs) are cyto- and bio-compatible [21]. They have attracted increasing attention in biomedical research due to its physicochemical stability, the potential for chemically modification, and ease of doping substances within its lumen, including therapeutic agents [22–25], enzymes [26], nucleic acid [27], and metal nanoparticles [28]. In addition, HNTs have been proven to enhance mechanical properties for numerous materials, such as alginate [26], chitosan [29] epoxy [30], nylon [25], rubber [31], and calcium phosphate [32]. Furthermore, HNTs have also been reported to chemotactically attract pre-osteoblasts [33] and enhance osteogenic differentiation [32,34,35].

Here, we used halloysite due to its known ability to improve polymer material properties and release bioactive agents in a sustained manner. HNTs were loaded with zinc nanoparticles. Zinc is one of the essential minerals that play an essential role in bone health. It affects multiple enzyme activities, collagen synthesis [36], and DNA synthesis [37], and it has been demonstrated to stimulate bone metabolism [38]. Zinc oxide nanoparticles are also a known and potent agent, that disintegrates bacterial cell membranes and accumulates in the cytoplasm leading to apoptotic cell death [39,40]. Therefore, zinc was selected as a coating for HNTs and then mixed with PLA for 3D printing. Fetal bovine serum (FBS) and NaOH were used to improve the surface hydrophilicity of a 3D printed scaffold. Scaffold mechanical properties and cell-material interactions were studied. We also coated the 3D printed scaffold with an antibiotic, gentamicin, to prevent contamination and assessed the drug efficiency after three weeks. This study aims to generate a 3D printed scaffold to support

bone regeneration and prevent bacterial contamination, which may be potentially used for bone defect therapy in the clinic.

## 2. Materials and Methods

### 2.1. Zinc Loaded into HNTs

Zinc nanoparticles (NPs) were deposited on the HNT surface by thermal decomposition of the metal acetate, as depicted in Figure 1. Zinc oxide (ZnO) reacted with acetic acid at 50 °C with continuous stirring, then the mixture was heated to a boil, and the reaction continued for 12 h, with additional acetic acid added during this time period. The resulting zinc acetate (Zn (OAc)$_2$) was filtered using Whatman #1 filter paper [40]. Then, 20 g of Zn (OAc)$_2$ mixed with 10 g of HNTs in 50 mL DI water and stirred for 12 h. After centrifugation, the pellet was collected and heated at 350 °C for 2 h, which led to thermal decomposition of the metal acetate on HNTs surface (ZnO-HNTs) [41].

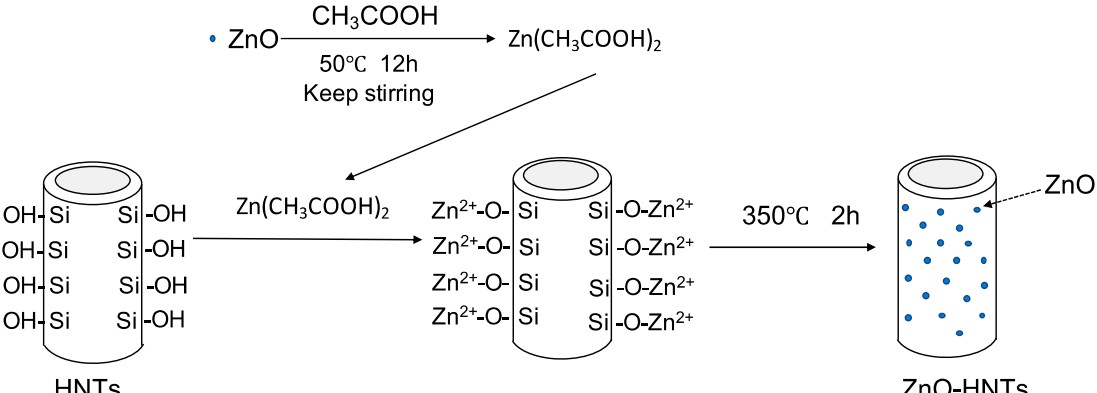

**Figure 1.** Facile synthesis and characterization of zinc oxide (ZnO) nanoparticles grown on halloysite nanotubes for enhanced photocatalytic properties.

### 2.2. Material Preparation

Four composition were tested in this study: PLA, PLA+HNTs, PLA+HNTs/Zn, and PLA+HNTs/Zn+gentamicin. These groups were printed using an ENDER 3 printer with similar setting; however, different filament compositions were used. Filaments were extruded using a Noztek Pro Extruder (Nortek Holdings Inc, West Sussex, England, UK) with a uniform diameter 1.75 ± 0.05 mm, but there was slightly different in filaments preparation for each group. For the PLA group, PLA filaments were extruded at 175 °C. For the PLA+HNTs group, in order to archive a uniform distribution of HNTs in PLA, 10 µL of silicon oil was added into 20 g PLA and vortexed for 10 min, then 1.2 g of HNTs were added and continually vortexed for another 10 min. Then mixture of PLA+HNTs were extruded at 170 °C. Filaments of PLA+HNTs/Zn prepared similarly as PLA+HNTs; the only difference is HNTs were loaded with Zn (30% w/w) and extruded at 165 °C. PLA+HNTs/Zn+gentamicin scaffolds were printed with PLA+HNTs/Zn filaments, and then they dipped into a 100 mg/mL gentamicin solution for 24 h.

### 2.3. D Printing

All filaments types were 3D printed into a pre-designed structure (squares) using an ENDER 3 printer at 225 °C. The squares were designed to be 6 × 6 × 2 mm with a pore size of 0.6 mm (Figure 2). The diameter of inside lattice supports was 0.6 mm.

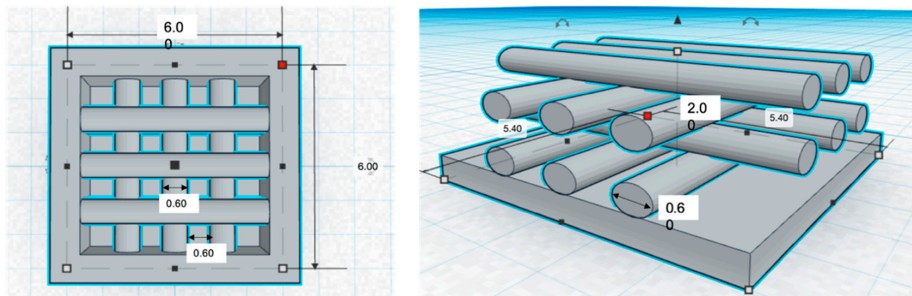

**Figure 2.** Computer aided design (CAD) drawing of the three dimensional (3D) printing square and design specifications.

## 2.4. Porosity

The porosity of the 3D printed disks was calculated through liquid displacement. One 3D square was immersed into 1.0 mL (*V1*) of DI water, then a series of vortexing and sonication was applied to force the liquid into the pores. The total volume of square and DI water was measured (*V2*), after the water was removed, the square and the remaining volume of DI water was measured (*V3*). The final porosity of the square was calculated as below:

$$\text{porosity} = \frac{V1 - V3}{V2 - V3}$$

## 2.5. Compression Testing

A Univert CellScale Testing device (Waterloo, Ontario, Canada) was used for compression test of the printed squares. The 3D printed squares were compressed at a speed of 10 mm/min with a 200 N load cell. The strain and stress profiles were recorded. A minimum of three tests were performed for each composition.

## 2.6. Surface Treatment of 3D Printed Square

According to the pilot study (Supplementary information), a sandwich coating (Figure 3) on the 3D printed squares was shown to significantly improve the surface hydrophilicity and facilitate cell adhesion. Therefore, we coated the 3D printed disk for three layers. Before coating, each disk was sterilized by immerging in 75% isopropanol for 10 min and air dried in cell culture hood. For the first layer, each square was immerged in fetal bovine serum (FBS) for 24 h; then each square was immerged into 10 N NaOH for 30 min and washed three times with sterilized DI water; for the last layer, squares were incubated in FBS again for 24 h. Squares with a three-layer sandwich-like coating were labeled as FBS+NaOH+FBS.

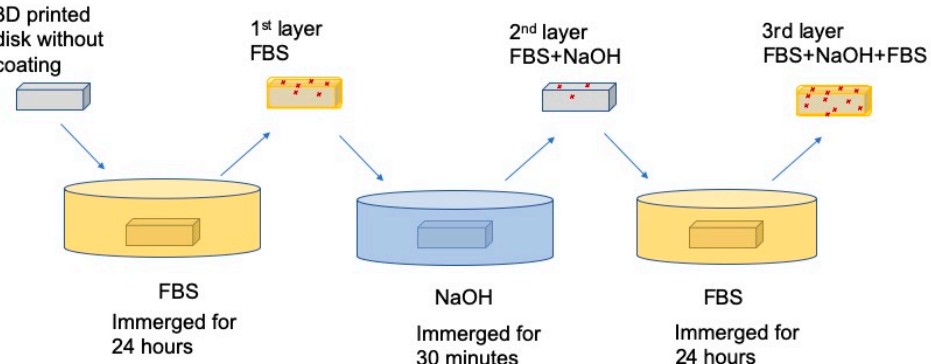

**Figure 3.** The process of applying the three-layered coating onto 3D printed squares.

### 2.7. Morphology and Surface Characterization

The morphology of 3D printed disks was observed using a scanning electron microscope (SEM) and laser confocal microscope. The distribution of HNTs and HNTs/Zn in the PLA filament was observed by an energy dispersive spectrometer (EDS) using a PLA filament in cross-section. The presence and nature of the surface coating was also determined by EDS.

### 2.8. Cell Metabolism

Surface modified squares were put into 48 wells plate, each well had one square and seeded with pre-osteoblast (MC3T3-E1, ATCC, Manassas, VA, USA) at a cell density of $1 \times 10^5$/well. Then, the cells were cultured in alpha modification of Eagle's medium ($\alpha$-MEM, Hyclone, GE Life Sciences, Marlborough, MA, USA) with 10% fetal bovine serum (FBS) and 1% Pen/Strep antibiotic (Life Technologies Corporation, Carlsbad, CA, USA) in a humidified incubator at 37 °C and 5% $CO_2$. Cells cultured with 3D printed scaffolds for 7, 14, and 21 days. An MTS (BioVision, Milpitas, CA, USA) was used to assess cell metabolism. A stock solution (40 μL) were added to each well and incubated at 37 °C in darkness for 2 h. After then, 200 μL of supernatant was taken to measure absorbance value at wavelength of 490 nm.

### 2.9. Mineralization-Alizarin Red Staining

Matrix mineralization was assessed through Alizarin Red S (ARS) staining. Cells cultured on the 3D printed squares for different time period (for 7, 14 and 21 days) were fixed with 4% paraformaldehyde for 15 min at room temperature, then stained with 2% ARS for 30 min. Then, all squares were washed by deionized water 4 times and observed under an Olympus BX41 light microscope. Cells cultured in monolayer were used as control.

### 2.10. Picrosirius Red Staining

Picrosirius Red is a specific collagen fiber stain that is capable of detecting thin collagenous fibers. The media was aspirated from the cell culture plates, and each culture well was washed with DPBS before being fixed in 4% paraformaldehyde. Fixed cells were stained with Picrosirius Red to quantify the amount of collagen secreted. Picrosirius stain was added to each well and removed after an hour incubation at room temperature. The cells were rinsed with 0.5% acetic acid solution twice and absolute alcohol twice. Digital images of stained squares were acquired using a brightfield microscope. Cells cultured in monolayer were used as control.

### 2.11. Antibacterial Efficiency

The antibacterial ability of PLA+HNTs/Zn+gentamicin squares against *Staphylococcus aureus* (*S. aureus*) was assessed. *S. aureus* was obtained as a gift from the laboratory of Dr. Rebecca Girono. The 3D printed squares were placed in 24-well cell culture plate and each well was inoculated with 1 mL *S. aureus* ($0.3 \times 10^7$ CFU/mL). The *S. aureus* was sub-cultured from a single colony and maintained in Muller Hinton broth. The plate was incubated in horizontal orbital microplate shaker at 37 °C for 12 h. The absorbance of the incubation solution at 630 nm was measured. Muller Hinton broth without PLA+HNTs/Zn+gentamicin squares and *S.aureus* set as negative and positive control, respectively.

### 2.12. Statistical Analysis

A one-way ANOVA or Student *t*-test was used for statistical analysis. Data were expressed as mean ± standard deviation. A *p*-value less than 0.05 was considered statistically significant.

## 3. Results

### 3.1. Distribution of HNTs and Zinc Nanoparticles in the PLA Filament

Filaments used to print PLA+HNTs, and PLA+HNTs/Zn squares were prepared by mixing PLA with HNTs or zinc-coated HNTs (HNTs/Zn). In order to determine whether the HNTs or HNTs/Zn were distributed throughout the PLA, filament cross-sections were analyzed with EDS. In Figure 4, all pictures represent the same visual field but present different elements. The primary element of PLA is carbon (C), which is exhibited all over the screen. Silicon (Si) and aluminum (Al) are the two major elements of HNTs, according to the graph, they were well distributed in the PLA filament. Zinc nanoparticles were coated into HNTs with 30% w/w, its distribution was detected by EDS as well. According to the EDS analysis, HNTs and HNTs/Zn were well distributed throughout the PLA filament.

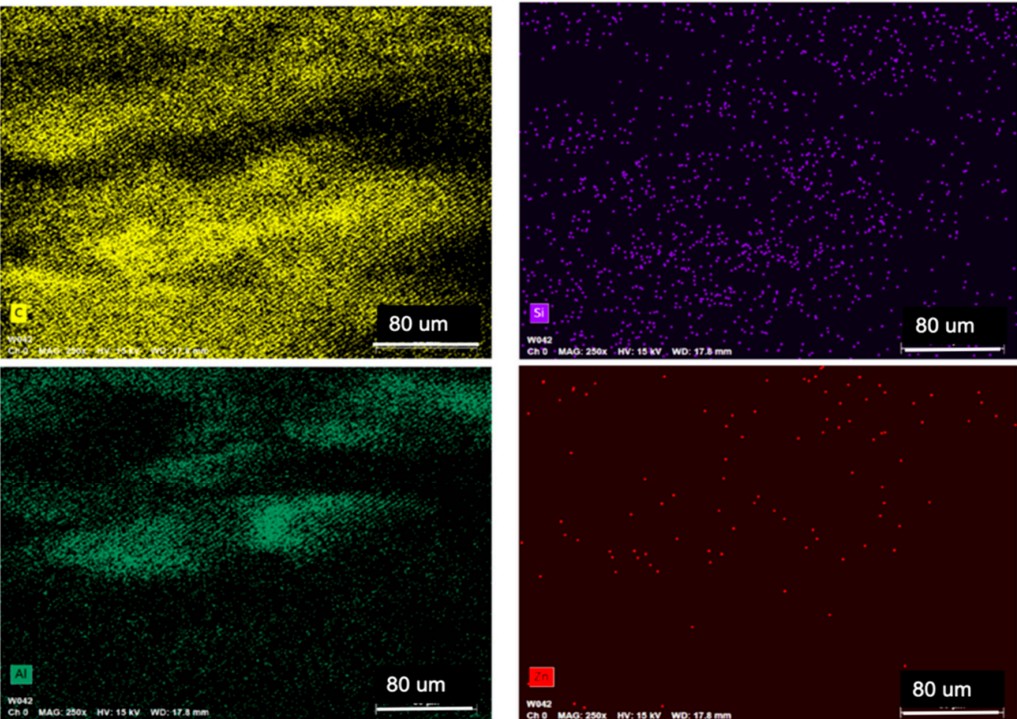

**Figure 4.** Energy dispersive spectrometer (EDS) analysis of polylactic acid and halloysite nanotubes (PLA+HNTs)/zinc (Zn) filament cross-section. All four pictures were focused on the same area, and each picture exhibits one element. Four elements were present. They are carbon (C), silicon (Si), aluminum (Al), and zinc (Zn).

### 3.2. Morphology of 3D Printed Squares and Their Surface Characteristics

All filaments were printed into a pre-designed square with a pore size of 600 μm × 600 μm and a layer height of 600 μm (Figure 2). Due to the limitations of the 3D printer used, the resolution changed slightly during printing. The exact pore size was determined using a laser confocal microscope (Figure 5). Based on the measurement of 60 pores from 20 different scaffolds, the average pore size of printed scaffolds is 584.16 ± 95.28 μm × 620.39 ± 93.03 μm and with a porosity of 60.22 ± 9.5%.

### 3.3. Compressive Strength

In order to evaluate the contribution of HNTs towards the enhancement of PLA's mechanical properties in the printed squares, the compressive strength of 3D printed scaffolds was analyzed, with and without HNTs addition. Scaffolds with HNTs (PLA+H and PLA+H+Zn) did have a higher strain percentage and higher average compressive modulus as compared to the squares without HNTs (PLA only), indicating that the addition of HNTs only contributed a slight but not significant

enhancement to the elasticity and compressive strength of PLA (Figure 6). Due to the limitation of the testing instrument, no scaffolds broke after the application of the maximum force (200 N). Therefore, we can't get the complete compression data. Based on the current data, there is a trendline in compression property enhancement from PLA to PLA+H+Zn, but this enhancement is not significantly.

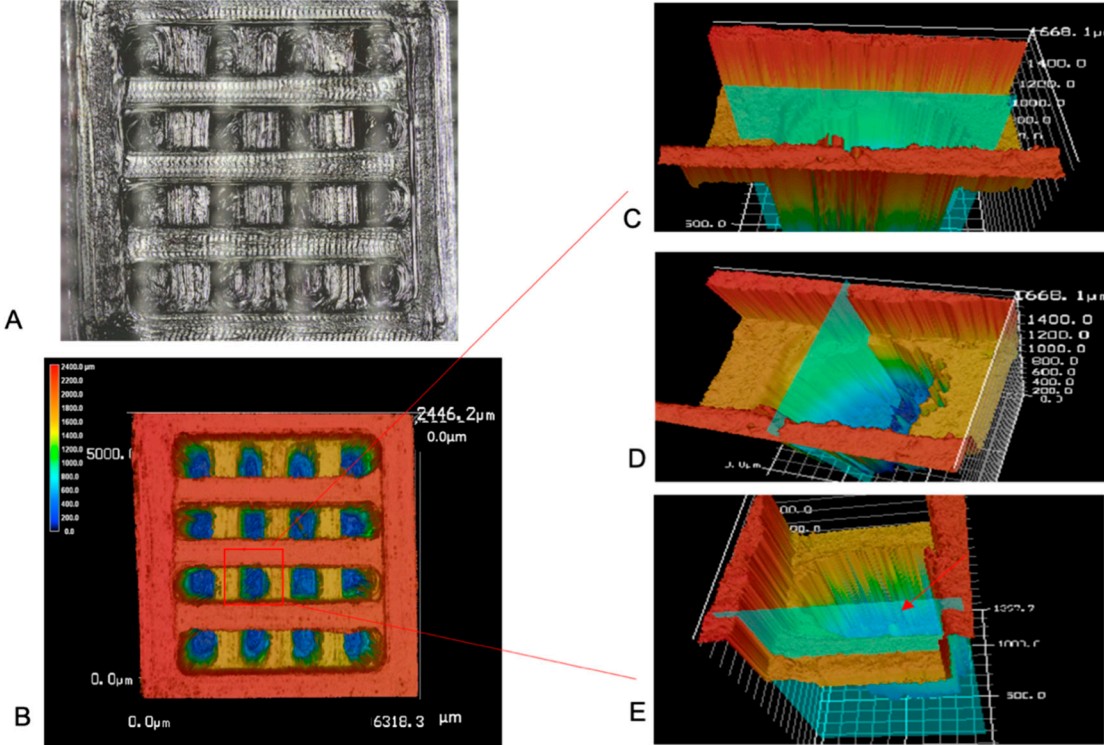

**Figure 5.** (**A**). Optical and laser combined picture of 3D printed square. (**B**). Laser confocal image of 3D printed square. (**C**). Horizontal section of selected pore, the horizontal distance was measured (584.16 ± 95.28 μm, n = 60). (**D**). Vertical section of selected pore, the vertical distance was measured (620.39 ± 93.03 μm, n = 60). (**E**). Vertical section of selected pore, the layer thickness was measured (423.15 ± 82.7 μm, n = 60).

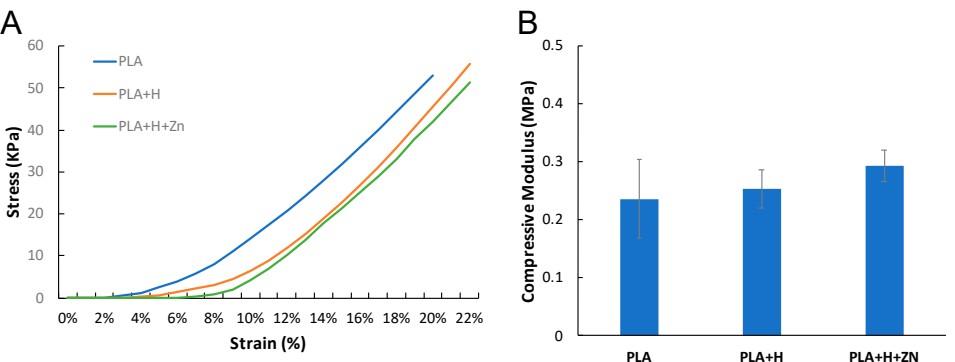

**Figure 6.** (**A**) Stress vs. strain profile and the compressive modulus of PLA, PLA+H, and PLA+H+Zn. (**B**) The compressive modulus: PLA = 0.24 ± 0.07 MPa, PLA+H = 0.25 ± 0.03 MPa, PLA+H+Zn = 0.29 ± 0.03 MPa (error bar with standard deviation, n = 5).

### 3.4. Chemical Deposition

After processing the sandwich-like layered surface modification, the hydrophilicity of the printed squares was significantly improved (Supplementary Figure S1). In our hypothesis, surface

hydrophilicity would keep increasing with each layer modification. However, the hydrophilicity decreased after the second layer modification was treated with NaOH, and then the hydrophilicity significantly increased after the third layer was added (Supplementary data, Figure S1). This phenomenon may have occurred because the NaOH eroded the chemicals that were deposited in the first layer modified with FBS. However, simultaneously, this erosion produced more links for chemical deposition, which lead to increased chemical deposition after the addition of the third layer (supplementary Figures S2 and S3, and Figure 7).

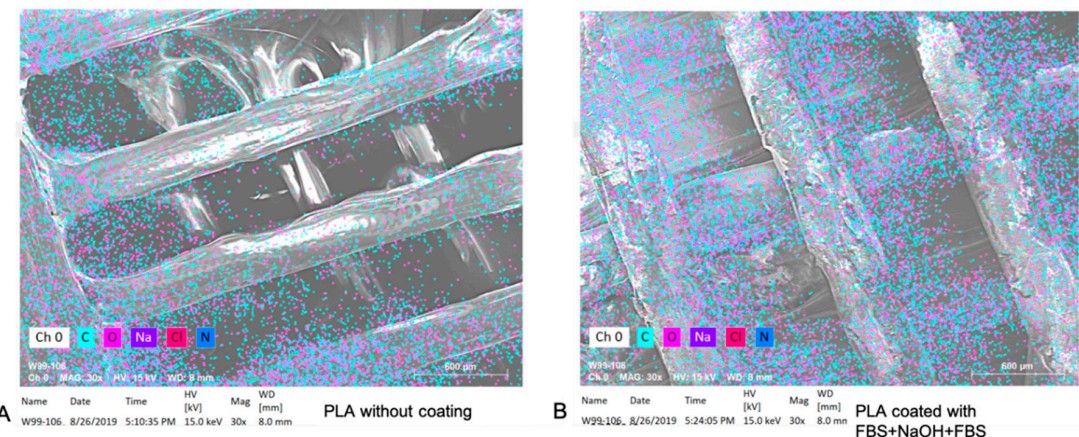

**Figure 7.** The EDS elemental analysis for PLA square with a naked surface (**A**) and a sandwich-like coating (**B**).

### 3.5. Antibacterial Studies

FBS contains many substances which may lead to the growth of undesired microorganism; therefore, we coated the PLA+H+Zn with gentamicin (PLA+H+Zn+G). Gentamicin is an efficient antibiotic against gram-positive and negative bacteria [42]. Even though they were stored at 37 °C for three weeks, they still efficiently inhibited bacterial growth (Figure 8). As expected, printed squares without gentamicin showed no bacterial growth inhibition.

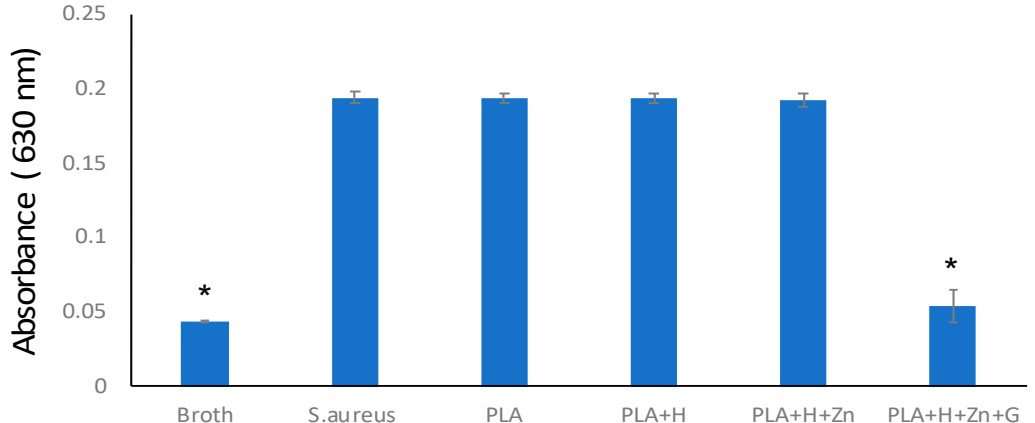

**Figure 8.** Bacterial growth inhibition. Broth without any samples or bacterial were set as the negative control (broth). *S. aureus* culture in broth was set as the positive control (*S. aureus*). The same amount of *S. aureus* suspension was co-cultured with a scaffold composed by PLA, PLA added with HNTs (PLA+H), PLA added with zinc-loaded HNTs (PLA+H+Zn), and PLA added with zinc-loaded HNTs and coated with gentamicin (PLA+H+Zn+G), (error bar with standard deviation, n = 3, * $p < 0.05$).

### 3.6. Response of Pre-Osteoblast to 3D Printed Squares

Many studies have shown that surface features, such as charge [43,44], roughness [45], adsorbed proteins [46], and hydrophilicity/hydrophobicity [47], greatly influenced cell attachment and subsequent cell behaviors. Our results consist with previous studies, cell adhesion was improved with hydrophilicity and increased protein attachment (Supplementary Data, Figure S4). In addition, cells preferred to proliferate on 3D squares as compared to the monolayer cultures (Figure 9).

The influence of gentamicin on cell metabolism was also assessed. Consistent with the study of Philip et al. [48], the presence of gentamicin did induce a small but transient effect on cell metabolism (Figure 9). However, the presence of gentamicin did not have a negative effect on mineralization (Figure 10).

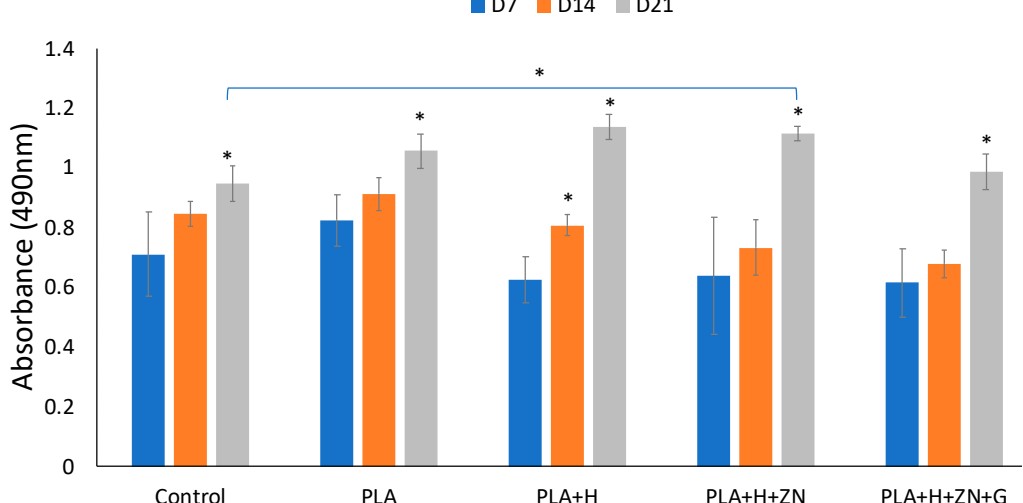

**Figure 9.** Cell metabolism in 21 days incubation. Cell cultured in monolayer culturing set as control. Then cells cultured with 3D squares consisted of 4 compositions: PLA, PLA added with HNTs (PLA+H), PLA added with zinc-loaded HNTs (PLA+H+Zn), and PLA added with zinc-loaded HNTs and coated with gentamicin (PLA+H+Zn+G). With one-way ANOVA analysis, there is a significant increase in cell metabolism of each group in 21 days. Comparing the cell metabolism in monolayer surface and 3D scaffold groups at day 21, cells cultured in PLA, PLA+H, and PLA+H+Zn had a significant higher cell metabolism than cells cultured in control group (error bar with standard deviation, n = 5, * $p < 0.05$).

Bone consists of bone cells and a mineralized collagenous matrix [49,50]. The main constituents of the bone matrix are hydroxyapatite ($Ca_{10}(PO_4)_6(OH)_2$) (50–70%) and an organic matrix (20–40%) [51]. Type I collagen is the major component of bone tissue extracellular matrix (ECM), which is mainly synthesized by osteoblasts. The synthesis of type I collagen is one of the markers of osteogenic differentiation [52]. Processed with Picrosirius Red staining, type III collagen stained red, and type I collagen stained yellow. In the first seven days since incubation, collagen secretion by cells in monolayer culture was negligible. In contrast, type III collagen synthesized by cells cultured on different 3D square compositions was very apparent (Supplementary Information, Figure S5).

Furthermore, compared to the inner space, more type III collagen was synthesized on the bottom surface of the square, and the transformation from type III collagen to type I collagen happened earlier on the bottom surface of the well (Supplementary Data, Figure S5 vs. Figure S6). Osteoblast differentiation in 3D scaffolds usually spreads from the scaffold periphery and gradually proceeds into the inner scaffold space [50]. Similar to the results seen with collagen synthesis, after 21 days of incubation, there was increased calcium deposition in the 3D scaffolds as compared to cells in monolayer culture (Figure 10). Calcium deposition indicates mineralization of the bone matrix, which is another marker for bone tissue formation. Alizarin Red S stained the calcium deposited in the collagenous matrix (red),

which was rarely found in monolayer culture after seven days of incubation (Supplementary Data, Figure S7).

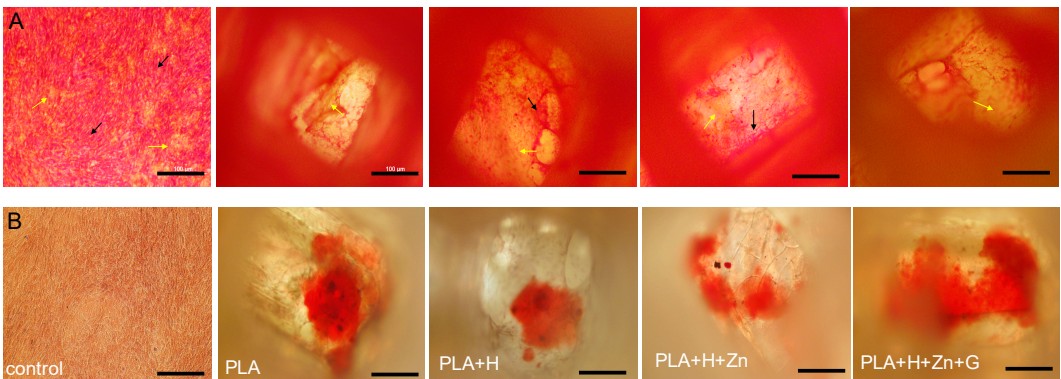

**Figure 10.** (**A**) Picrosirius Red Stain for bottom layer of 3D squares after 21 days incubation. The red color represents type III collagen (black arrows) and yellow color represents type I collagen (yellow arrows). Cells incubated in monolayer set as control (fare left figures). (**B**) Alizarin Red stain for calcium deposition after 21 days incubation (scale bar is 100 μm).

## 4. Discussion

PLA has been extensively studied as a candidate for drug delivery and tissue engineering due to its biocompatibility, biodegradability and good mechanical properties; however, hydrophilicity modification is necessary to decrease its hydrophobicity for in vitro and in vivo studies. Sodium hydroxide (NaOH) is a common chemical used in every lab. Its hydroxyl (-OH) groups can be introduced through hydrolysis, which cleaves the ester bonds. In addition, the introduced -OH groups can be used to bind bioactive molecules such as collagen [53], fibronectin [54], and arginine-glycine-aspartic acid (RGD) [55] to regulate cell adhesion.

In our early exploration of surface modification, we have tried other coating strategies, including coating of pure NaOH (10 N) and collagen. However, may due to the thickness of the filament or the high temperature processing, PLA has a certain degree of resistance to NaOH erosion, that results in a poor cell adhesion. In contrast, cells were flourishing on fibers that formed by collagen crosslinking. While collagen fibers forming networks in- and out-side of 3D scaffold, there is no optional way to precisely control the distribution and direction of these networks, nor of their pore size. The affection of collagen fibers on cell growth superseded the one brought by 3D scaffolds.

In this study, we used FBS as the ground and top layer and incorporated with NaOH to generate a sandwich coating (FBS+NaOH+FBS) on the PLA surface. Fetal bovine serum is a widely used serum supplement for in vitro cell culture. In the pilot study of FBS coating, cells successfully attached on scaffold surface, which indicates an enhancement both in hydrophilicity and cell compatibility, simultaneously, FBS did not introduce a morphology change. Following with a second layer treatment of NaOH, there is an attenuation in element deposition as well as in hydrophilicity. This phenomenon indicated a chemical erosion caused by NaOH, but chemical bond cleavage may also associate with remained free chemical bonds that are available for further deposition. The increased FBS deposition in third layer coating confirmed this hypothesis (Supplementary Information, Figures S2 and S3). The increase of hydrophilicity and cell attachments are also indirectly supported an additional deposition of chemicals on eroded chemical bonds (Supplementary Information, Figures S1 and S4). Due to the prominent enhancement on hydrophilicity and bio-friendliness, we utilized the three-layer coating strategy in this study. This deposition-erosion-deposition strategy may be also appropriate for other biomaterials' surface modification.

With the developing of 3D printing, there are several techniques used to fabricate 3D structure. The one used in this study is fused-deposition modeling (FDM). This method of printing deposits melted thermoplastic in thin layers and laydown at the designed location associated with CAD

model. Therefore, FDM can be used to fabricate complicated 3D structure. However, due to the high extruding temperature required by biomaterials, such as the extruding temperature for PLA is 225 °C, no bio-factors nor cells can be printed with biomaterials. This is the biggest limitation of FDM during tissue engineering.

## 5. Conclusions

In this study, we 3D printed squares composed of PLA and zinc doped HNTs for use as a potential bone implant. The material's porosity mimicked that of human bone tissue. When a unique sandwich coating of FBS+NaOH+FBS was applied to the printed squares, hydrophilicity was enhanced that facilitated cell adhesion and metabolism. Sandwich-coated PLA squares were also osteoinductive as seeded pre-osteoblasts differentiated into osteoblasts without the addition of exogenous osteogenesis agents. In addition, an external coating of gentamicin reduced the risk of infection without negatively influencing osteogenesis. The newly designed hybrid material, PLA+H+Zn, also possessed good mechanical strength and osteoinductivity and may serve as a candidate for 3D printing of bone implants. Furthermore, the surface modification strategy used in this study may also be used for other 3D printing applications.

**Supplementary Materials:** The following are available online at http://www.mdpi.com/2076-3417/10/11/3971/s1, Figure S1: Contact angle at each layer modification; Figure S2: The SEM images and EDS element analysis for 3D printed scaffold; Figure S3: Deposition of Na for each step modification.; Figure S4: Cell adhesion on surface that with each layer modification; Figure S5: Picro-sirius Red Stain for type I (yellow) and type III (red) collagen that are synthesized at bottom layer of 3D scaffolds with different time incubation. Arrows point to type I collagen; Figure S6: Picro-sirius Red Stain for type I (yellow) and type III (red) collagen that are synthesized at inner space of 3D scaffolds with different time incubation. Arrows point to type I collagen; Figure S7: ARS stain for calcium deposition with cells cultured in regular 2D environment (control) and 3D scaffolds. The red color represents calcium deposition.

**Author Contributions:** Conceptualization, Y.L., D.K.M.; Methodology Y.L., A.H.; Data analysis, Y.L., A.H., D.K.M.; writing, Y.L., D.K.M. All authors have read and agreed to the published version of the manuscript.

**Funding:** This research was funded by a grant (to DKM) from the Louisiana Biomedical Research Network (through an Institutional Development Award (IDeA) from the National Institute of General Medical Sciences of the National Institutes of Health under grant number P20 GM103424-17. Funding support was also received from the College of Applied and Natural Sciences (Louisiana Tech University) Matching Grant Program.

**Acknowledgments:** This research was funded by a grant (to DKM) from the Louisiana Biomedical Research Network (through an Institutional Development Award (IDeA) from the National Institute of General Medical Sciences of the National Institutes of Health under grant number P20 GM103424-17. Funding support was also recevied from the College of Applied and Natural Sciences (Louisiana Tech University) Matching Grant Program.

**Conflicts of Interest:** The authors declare no conflict of interest.

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
