# Peer review of "Surface Modification of 3D Printed PLA/Halloysite Composite Scaffolds with Antibacterial and Osteogenic Capabilities"

_applsci, doi:10.3390/app10113971_

Round 1
Reviewer 1 Report
The manuscript no. 812517 entitled: „ Zinc Doped Halloysite Cooperating with Polylactic Acid for Bone Regeneration” presents synthesis and in vitro evaluation of new biomaterials dedicated for bone tissue engineering.
The article is very interesting and will be appropriate to publish in Applied Science journal, when some minor remarks will be considered:
- Materials and Methods
2.3. D Printing
- a) line 114: Should be “3D Printing”?
- b) lines 116-117: Please, correct reference. It is : „Error! Reference source not found”
2.8. Cell proliferation
- a) How the authors sterilized the obtained biomaterials?
- b) The information about time of incubation should be added. Is it 7, 14, and 21-days?
2.9. Mineralization-Alizarin Red Staining
- a) The information about time of incubation should be added. Is it 7, 14, and 21-days?
- Results and Discussion
3.5. Antibacterial Studies
Line 223: Please, change „41” as number of reference.
3.6. Response of Pre-Osteoblast to 3D Printed Squares
The authors precisely indicated the factors, which affect cell behavior and they added appropriate references. Nevertheless, in the case of factor - „adsorbed proteins”, there is no reference. Please add suitable reference. For example: Klimek K., Ginalska G. ”Proteins and Peptides as Important Modifiers of Polymer Scaffolds for Tissue Engineering Applications – A Review”. Polymers 2020, 6, E844.
Author Response
Our response to the issues and concerns expressed in your review is included in our attachment.

Reviewer 2 Report
The article entitled "Zinc Doped Halloysite Cooperating with Polylactic Acid for Bone Regeneration ". This is an average article with serious flaw and need serious revision before publication or rejected. I will recommend it for Rejection. The reasons for rejection are bellow:
- This paper is based on 3D printed scaffolds for osteogenic differentiation but title does not represent the work. Thus, title should be changed.
- The authors have used halloysite nanotubes (HNTs) in the polymer matrix to improve the mechanical and biological properties, but there is no data related to HNTs. Why?
- Why zinc-loaded HNTs were used for 3D printing? What is the role of Zinc?
- 3, Why authors have used three-layered coatings onto 3D printed sample? Does NaOH affect the FBS functionality?
- Why there is no SEM imaged of 3D printed sample showing the different surfaces of PLA, PLA+H, PLA_H+Zn?
- Figure 6, mechanical characterizations, why there is no scale on x- and y-axis?
- The stress-strain curve is not complete (final breakdown)? How can be sure about valu of compressive strength around 200-300 KPa?
- “Scaffolds with HNTs (PLA+H andPLA+H+Zn) did have a higher strain percentage and higher average compressive modulus ascompared to the squares without HNTs (PLA only), indicating that the addition of HNTs onlycontributed a slight enhancement to the elasticity and compressive strength of PLA”. Is this change statistically significant or not?
- Mechanical test should be analyzed properly. The stress-strain curve of PLA+H and PLA+H+Zn samples does not show significant inclined with respect to each other but compressive modulus is showing difference, why? What is the role of Zn functionalization?
- Figure 8, the bacterial growth inhibition results for PLA+H and PLA+H+Zn are almost same. What does it mean? Why did you used Zn?
Author Response

(The authors gave the same response as above.)

Reviewer 3 Report
I would like to thank the authors for the interesting submission on the development of 3D printed zinc-loaded HNTs for bone tissue engineering. The results are interesting but certain issues need to be resolved to guarantee publication:
1. line 31: only a minority of orthopaedic procedures require bone grafting. Please rephrase to make it clear
2. Additional limitations exist for the use of autografts such as their limited availability and the need for an extra incision which carries extra risks including donor site morbidity etc. Please be accurate and include either all or none of the limitations of autografts.
3. Line 45: you transition abruptly from the need for new methods of making bone grafts to 3D printing. Are there any other methods of fabricating artificial bone grafts?
4. One of the major limitations of this method is that the high extrusion temperatures do not allow simultaneous printing of cells. Please discuss this limitation in your discussion and compare to similar literature
5. Better explanation is needed for the rationale of using FBS and NaOH. Please be more elaborate for your readers.
6. Line 150: MTS does not assess proliferation on biomaterials even though many papers erroneously use it for this purpose. Materials can alter cell metabolism (see Klontzas et al. Acta Biomaterialia 2019) and therefore MTS can assess metabolic activity but not proliferation. In order to judge proliferation Live/Dead assays in conjunction with protein or DNA content measurements can be used. Please rephrase and discuss this in your discussion as a limitation.
7. For the ARS please include a control where scaffolds without cells have been also stained. It is not unheard of that materials stain with ARS without the presence of cells. Also, include a graph with quantification of the staining.
8. Osteogenic differentiation is favoured by scaffolds with a pore size of 100-400 μm. Why did you select 600?
9. Can you modify with peptides? please explain in the text
10. line 86: Gentamycin is not an antibody. Please correct.
11. Why did you specifically select gentamicin as an antibiotic agent? It is nowadays rarely used in clinical practice. Another agent active against MRSA would be more useful than gentamycin
12. Extrusion at 170oC may damage the antibiotic. Is there any data proving otherwise?
13. Line 177: Please change standard errors to standard deviations which better represent the whole population and not only the mean. Standard errors usually look better but provide limited information in such studies.
14. Figure 6: Figure 6a is squeezed. Please correct. Is there any statistically significant difference between the results of figure 6b?
15. In your EDS analysis elements seem to exist in the void space. Why?
16. Figure 9: It looks squeezed. Please correct. Also it is not clear whether any statistical differences exist between groups and between time points. Please provide a two-way anova analysis.
17. Most of the results presented in figure 10 look like false positive staining because of insufficient washout. When staining with ARS the samples need to be washed under tap water until no more stain can be removed from the sample. Please repeat and add quantification results which will be more convincing.
Author Response

(The authors gave the same response as above.)

Round 2
Reviewer 2 Report
The manuscript has been significantly improved. Most of the comments have been replied with satisfactory answer. Therefore, I change my previous decision from Rejection to Acceptance. I have no concern and this manuscript can be accepted for publication.
Reviewer 3 Report
I am satisfied with the corrections